# Dimethyl sulfate and diisopropyl sulfate as practical and versatile *O*-sulfation reagents

Shuaishuai Yue[1,4], Guoping Ding[2,3,4], Ye Zheng[1], Chunlan Song[1] ✉, Peng Xu [2] ✉, Biao Yu [2] & Jiakun Li [1] ✉

*O*-Sulfation is a vital post-translational modification in bioactive molecules, yet there are significant challenges with their synthesis. Dialkyl sulfates, such as dimethyl sulfate and diisopropyl sulfate are commonly used as alkylation agents in alkaline conditions, and result in the formation of sulfate byproducts. We report herein a general and robust approach to *O*-sulfation by harnessing the tunable reactivity of dimethyl sulfate or diisopropyl sulfate under tetra-butylammonium bisulfate activation. The versatility of this *O*-sulfation protocol is interrogated with a diverse range of alcohols, phenols and *N*-OH compounds, including carbohydrates, amino acids and natural products. The enhanced electrophilicity of the sulfur atom in dialkyl sulfates, facilitated by the interaction with bisulfate anion ($HSO_4^-$), accounts for this pioneering chemical reactivity. We envision that our method will be useful for application in the comprehension of biological functions and discovery of drugs.

*O*-Sulfation (also known as sulfurylation) is a crucial post-translational modification with significant implications in developmental biology, immunology, and neurobiology, as well as disease processes such as cancer, inflammation, and central nervous system disorders[1-4]. This modification can occur in a diverse range of biomolecules, including polysaccharides, peptides, proteins, natural products, and drug metabolites (Fig. 1A)[5-7]. The unique nature of sulfate group can introduce both specific and non-specific recognition via an electrostatic or a hydrogen bonding interaction, posing challenges in modulating sulfate-protein interactions[8-10]. Sulfation of appropriately designed, non-natural scaffolds expands their structural diversity and provides valuable insights into a wide range of biological functions[11,12]. Notably, tyrosine sulfate isosteres of CCR5 have been designed as a tool to comprehend HIV-1 entry mechanism[13,14]. However, one of limiting factors in studying the structure-activity relationships of sulfate modifications in biology is their rather difficult availability[15,16].

While *O*-sulfation appears to be a one-step reaction, chemical synthesis of sulfated molecules remains challenging due to their lability to acid and temperature, insolubility in nearly all organic solvents,

and limited maneuverability following introduction[17,18]. As a result, extensive studies and synthetic pathways to organic sulfates are constantly being sought[19-22]. Sulfation based on $H_2SO_4$ has been successfully applied to simple alkenes and alcohols, but it is unsuitable for compounds containing delicate functional groups, primarily due to the strong acidity of sulfuric acid[23-26]. Sulfur trioxide-nitrogen base complexes are the most commonly used reagents for sulfating various molecular motifs containing alcoholic, phenolic, amino, thiol and other functional groups[27-30]. Conversely, the synthetic pathway for sulfur trioxide-nitrogen base complexes often necessitates abrasive and severe conditions, which ultimately constrain their synthetic adaptability. An alternative strategy to access the target scaffold in a masked sulfate diesters is the so-called early-stage sulfation, which was deblocked to afford *O*-sulfate in the final step[31-37]. It is complementary to the direct sulfation as the masked sulfate diesters are uncharged, can be usually purified with traditional method, and are stable to the subsequent transformations. Various protecting groups of sulfate diesters were explored in early-stage *O*-sulfation, such as phenyl and its variants[31], neopentyl (*n*P)[32], isobutyl (*i*Bu)[32], 2,2,2-trichloroethyl

[1]State Key Laboratory of Chemo/Biosensing and Chemometrics, College of Chemistry and Chemical Engineering, Hunan University, Changsha 410082, P. R. China. [2]State Key Laboratory of Chemical Biology, Shanghai Institute of Organic Chemistry, Chinese Academy of Sciences, 345 Lingling Road, Shanghai 200032, P. R. China. [3]Key Laboratory of Structure-based Drug Design & Discovery (Ministry of Education), School of Pharmaceutical Engineering, Shenyang Pharmaceutical University, Shenyang 110016, P. R. China. [4]These authors contributed equally: Shuaishuai Yue, Guoping Ding. ✉e-mail: songcl@hnu.edu.cn; peterxu@sioc.ac.cn; jkli@hnu.edu.cn

Article

**Fig. 1 | Significance of *O*-sulfation and dimethyl sulfate as reagent for late-stage *O*-functionalization. A** Representative drug molecules and natural products containing organosulfates. **B** Dimethyl sulfate as reagent for late-stage *O*-

functionalization. **C** Nucleophilic substitution on DMS by $B_{AL}2$ and $A_{SU}2$ mechanisms. DMS, dimethyl sulfate.

## Table 1 | Investigation of the reaction conditions[a]

| Entry | Variation from standard conditions[a] | Yield of 1 (%)[b] |
|---|---|---|
| 1 | Standard conditions | 84 |
| 2 | Without DMS | n.d. |
| 3 | Without Bu$_4$NHSO$_4$ | n.d. |
| 4 | Bu$_4$NBF$_4$ instead of Bu$_4$NHSO$_4$ | 21 |
| 5 | Bu$_4$NOAc instead of Bu$_4$NHSO$_4$ | 58 |
| 6 | Bu$_4$NI instead of Bu$_4$NHSO$_4$ | 49 |
| 7 | KHSO$_4$ instead of Bu$_4$NHSO$_4$ | n.d. |
| 8 | NH$_4$HSO$_4$ instead of Bu$_4$NHSO$_4$ | 28[c] |
| 9 | NaH (5.0 equiv) | 78[d] |
| 10 | 50 °C | 41 |
| 11 | 100 °C | 65 |
| 12 | Under air | 61 |

n.d., no detected.

[a]Standard conditions: reactions were performed with 3-phthalimido-1-propanol **1a** (0.2 mmol, 1.0 equiv), dimethyl sulfate (DMS, 1.2 equiv) and Bu$_4$NHSO$_4$ (1.2 equiv) in CH$_3$CN (1 mL) under argon at 80 °C for 12 h.

[b]NMR yields were determined by using CH$_2$Br$_2$ as an internal standard.

[c]Ammonium sulfate as the product.

[d]The product was obtained from *O*-methylation.

(TCE)[33–36], trifluoroethylene (TFE) groups[37]. In addition, the sulfitylation-oxidation protocol[38] and sulfur (VI) fluoride exchange reaction (SuFEx)[31,39,40] could also be used to generate sulfate diesters, making them powerful tools for early-stage sulfation on complex peptides and carbohydrates.

In term of structure components, dialkyl sulfates, such as dimethyl sulfate (DMS) and diisopropyl sulfate (DPS) are typically classified as sulfate diesters[41–43]. They consist of a sulfate group with two alkyl groups attached to it. This specific feature makes them highly effective as alkylation agents under alkaline conditions by a $B_{AL}2$ (bimolecular, base-promoted, alkyl cleavage, nucleophilic substitution) mechanism

(Fig. 1C-i). The sulfate group then acts as a leaving group, resulting in the formation of innocuous sulfate byproducts finally. Inspired by the tunable reactivity of dimethyl carbonate (DMC) for methylation and methoxycarbonylation[44], we envisioned that a different activation method to increase the electrophilicity of sulfate group in dialkyl sulfates, which may generate alkyl sulfate monoester **B** through an $A_{SU}2$ (bimolecular, acid-promoted, sulfate cleavage, nucleophilic substitution) mechanism (Fig. 1C-ii). This chemical process facilitates the transfer of sulfate groups to target molecules, but it presents significant challenges. A suitable activation system for dialkyl sulfates should also function as the reagent for removal of the alkyl group

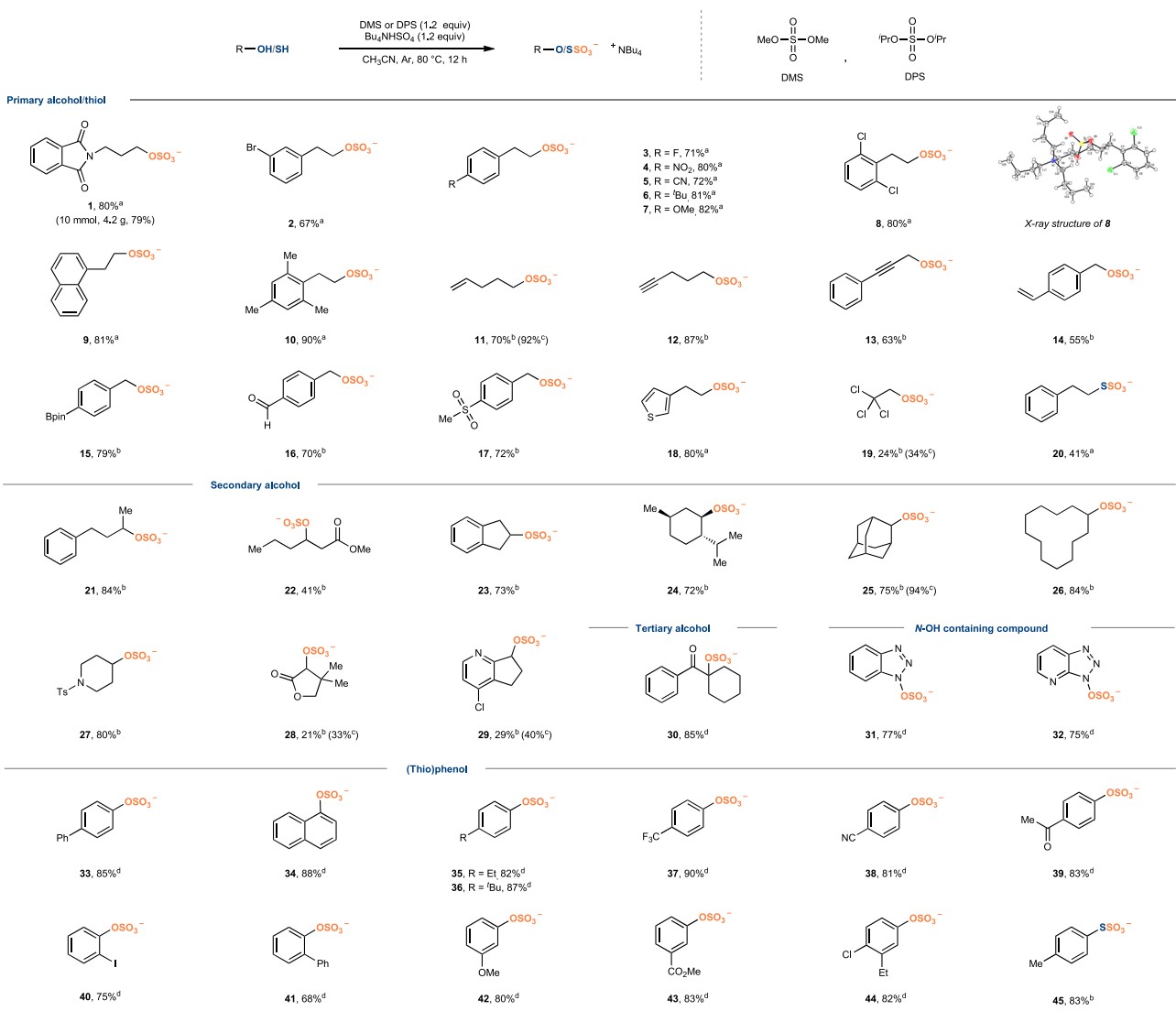

**Fig. 2 | Substrate scope of *O/S*-sulfation.** [a]Reaction conditions a as shown in Table 1. [b]DPS (1.2 equiv) and Bu₄NHSO₄ (2.0 equiv) were used in reaction conditions b. [c]NMR yields were determined by using CH₂Br₂ as an internal standard. [d]Na₂S₂O₇ (2.0 equiv) and Bu₄NHSO₄ (1.1 equiv) were used. Note: The liquid DMS and DPS are volatile and toxic. Exercise extreme caution when handling the liquid. DPS, diisopropyl sulfate.

(methyl or isopropyl) in sulfate monoester **B** as shown in Fig. 1C, to release the final sulfate product **C**. Alkyl protected sulfate ester **B** is always stable, a selective deprotection strategy is not easy to differentiate the fission rate of the two S–O bonds[45]. Despite a comprehensive study for removal of various protecting groups in sulfate monoesters[31–37], methyl and isopropyl remains elusive. Moreover, the design of activation system should be compatible with acid-labile sulfate **C**, avoiding its decomposition. Herein, we demonstrated a strategy that utilizes dialkyl sulfates (DMS and DPS) as source of sulfates to achieve *O*-sulfation under tetrabutylammonium bisulfate activation.

## Results

After an extensive optimization campaign based on our working hypothesis, we identified DMS as the optimal sulfate source and Bu₄NHSO₄ as the ideal activation agent. The reaction between the initial model substrate **1a** and DMS (1.2 equiv) under mildly acid conditions (Bu₄NHSO₄, 1.2 equiv) in CH₃CN at 80 °C for 12 h, provided the desired tetrabutylammonium 3-phthalimido-1-propanol sulfate **1** in 84% yield (Table 1, entry 1). Control experiments revealed that DMS and Bu₄NHSO₄ were both essential for this sulfation (Table 1, entries 2,

3). Without DMS or Bu₄NHSO₄, the reaction did not proceed at all. The use of Bu₄NBF₄, Bu₄NOAc, or Bu₄NI as additive that could potentially release their conjugated acids from the equilibrium with **1a**, resulted in lower yield (Table 1, entries 4–6). Similarly, the shift from tetrabutylammonium to other cations was ineffective for this transformation (Table 1, entries 7, 8). These outcome confirmed the crucial role of Bu₄NHSO₄ for enhanced solubility of sulfate product **1** as tetrabutylammonium salt as well as facile removal of the methyl group in sulfate monoester **B**[46]. Not surprisingly, *O*-methylation rather than sulfation occurred in the presence of sodium hydride (Table 1, entry 9)[41–43]. When the reaction was conducted at lower or higher temperature, a notable decease of yield was observed (Table 1, entries 10, 11). Exposure to air is possible for *O*-sulfation (Table 1, entry 12), obviating the need for rigorous deoxygenation and dehumidification.

Studies of the scope of this *O*-sulfation are summarized in Fig. 2. The reaction performed well to a wide range of primary and secondary alcohol substrates. The sulfation of most primary alcohols proceeds effectively with DMS (conditions a), while DPS is more applicable to both primary and secondary alcohols (conditions b). A variety of functional groups are tolerated including amide (**1**), halides (**2, 3, 8**), nitrate (**4**), nitrile (**5**), ether (**7**), alkene (**11**), alkynes (**12, 13**), boronic

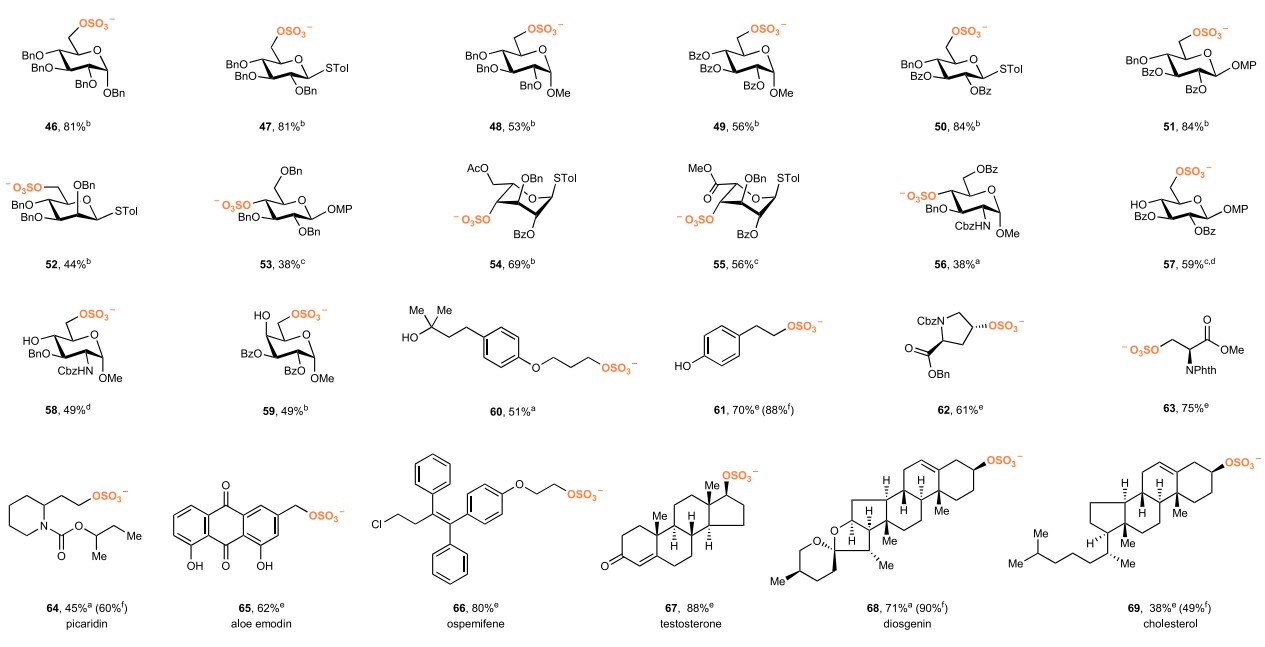

**Fig. 3 | Late-stage *O*-sulfation of carbohydrates, amino acids and natural products.** [a]Standard conditions as shown in Table 1. [b]DMS (2.0 equiv) and Bu$_4$NHSO$_4$ (2.0 equiv) were used. [c]24 h instead of 12 h. [d]Bu$_4$NHSO$_4$ (2.0 equiv) was used. [e]DPS (1.2 equiv) and Bu$_4$NHSO$_4$ (1.2 equiv) were used. [f]NMR yields were determined by using CH$_2$Br$_2$ as an internal standard.

ester (**15**), aldehyde (**16**), sulfone (**17**), esters (**22, 28**), ketone (**30**), sulfonamide (**27**), aromatic (**9**) and heterocycles (**18, 29**). Despite the likelihood of erosion under acid conditions, anisole (**7**), alkene (**11**), and alkynes (**12, 13**) remained intact. The week nucleophilic trichloroethanol was also a competent partner with acceptable yield of the target product (**19**). Both cyclic and acyclic secondary alcohols were efficiently transformed into the sulfates (**21**–**29**). The structural identity of the target sulfate group was unambiguously established by single crystal X-ray diffraction for compound **8**. The practicability of this approach was further demonstrated on the gram scale synthesis of **1**. Due to the lability of organic sulfates on chromatography, such as **11**, **19** and **25**, the purified compound yields were significantly lower than the yields determined by $^1$H NMR. We could also generalize the transformation beyond primary and secondary sulfates synthesis to include the formation of sulfates from phenols as well as tertiary alcohols and *N*–OH containing compounds. Their relative weak nucleophilicity compared to common alcohols[47] makes them sluggish to attack dialkyl sulfates, even with acid activation. The use of stronger electrophilic sodium pyrosulfate instead of DMS or DPS was crucial to obtain high yields. With this modified protocol, the bulky sulfated product (**30**) was obtained in 85% yield; *N*–OH containing compounds are suitable in this system to obtain *N*–OSO$_3$ products (**31**, **32**) in good yields. Moreover, the reaction performed well with electron-rich to electron-deficient phenols (**33**–**44**). This protocol was also successful in producing *S*-sulfation products **20** and **45** for thiol and thiophenol. However, nitrogen compounds such as amine and imine (Supplementary Table 4) did not yield any desired sulfation products, and all starting materials were recovered.

Hydroxyl groups are not only prevalent in large quantity as alcohols and phenols, but they are also present in a wide variety of complex natural products and drugs. The mild reaction conditions described in this method make it applicable to a large variety of carbohydrates, amino acids and steroids, as shown in Fig. 3. Various sugar derivatives with different protecting groups and glycosidic bonds were successfully sulfated under these conditions. Armed (**46**–**48**, **53**) and disarmed glucoside (**49**), mannoside (**52**), idose (**54**), iduronic acid (**55**)

and glucosamine derivatives (**56**) reacted smoothly to form the corresponding 6-*O* and 4-*O* sulfated products in yields of 38–81%. This method was also applied to molecules containing different hydroxyl moieties within the same structure, enabling site-selective *O*-sulfation among primary and secondary/tertiary hydroxyl groups in glycoside (**57**), glucosamine (**58**), galactoside (**59**) and **60**. The selectivity in these reactions appears to be mainly controlled by steric effects, as no distinction was observed among substrates with similar hindrance (Supplementary Fig. 7). Moreover, the selectivity profile of this method extends to differentiate between phenols and alcohols, as exemplified by the selective sulfation of a primary alcohol over a phenol in compound **61**. Amino acids, such as proline (**62**) and serine (**63**) were viable in this *O*-sulfation. Furthermore, we examined this sulfation reaction within various densely functionalized architectures. Notably, pharmaceuticals and repellents like picaridin (**64**), aloe emodin (**65**), and ospemifene (**66**) were amenable to this reaction. In addition, natural steroid hormone derivatives, such as testosterone (**67**), diosgenin (**68**), and cholesterol (**69**), were also competent substrates, allowing for the synthesis of desired products in high yields. These results highlight the broad utility of this sulfation method for late-stage modifications of bioactive molecules.

Next, a series of control experiments were carried out to obtain further insight into the reaction mechanism (Fig. 4). On-line reaction monitoring by NMR was performed to detect reaction intermediates and their reactivity (Fig. 4A). When the mixture of alcohol **1a** and DMS was activated with 1.2 equiv of Bu$_4$NHSO$_4$, NMR spectra showed generation of desired sulfate **1** is concomitantly accompanied by formation of methyl monosulfates **1-B** and methyl sulfate **D/D′** respectively. As the reaction progressed, these intermediate species were converted to some extent into the final sulfate product. Notably, the release of MeOH was also observed during the reaction. This results promoted us to investigate the additional role of these key intermediates in the sulfation. DMS reacts with tetrabutylammonium bisulfate (Bu$_4$NHSO$_4$) to yield a mixture of **D** and **D′** (Fig. 4B) rather than SO$_3$ complex, which substantiates the possible path b in the reaction (Fig. 1C-ii). The higher yield of **1**, in comparison to the

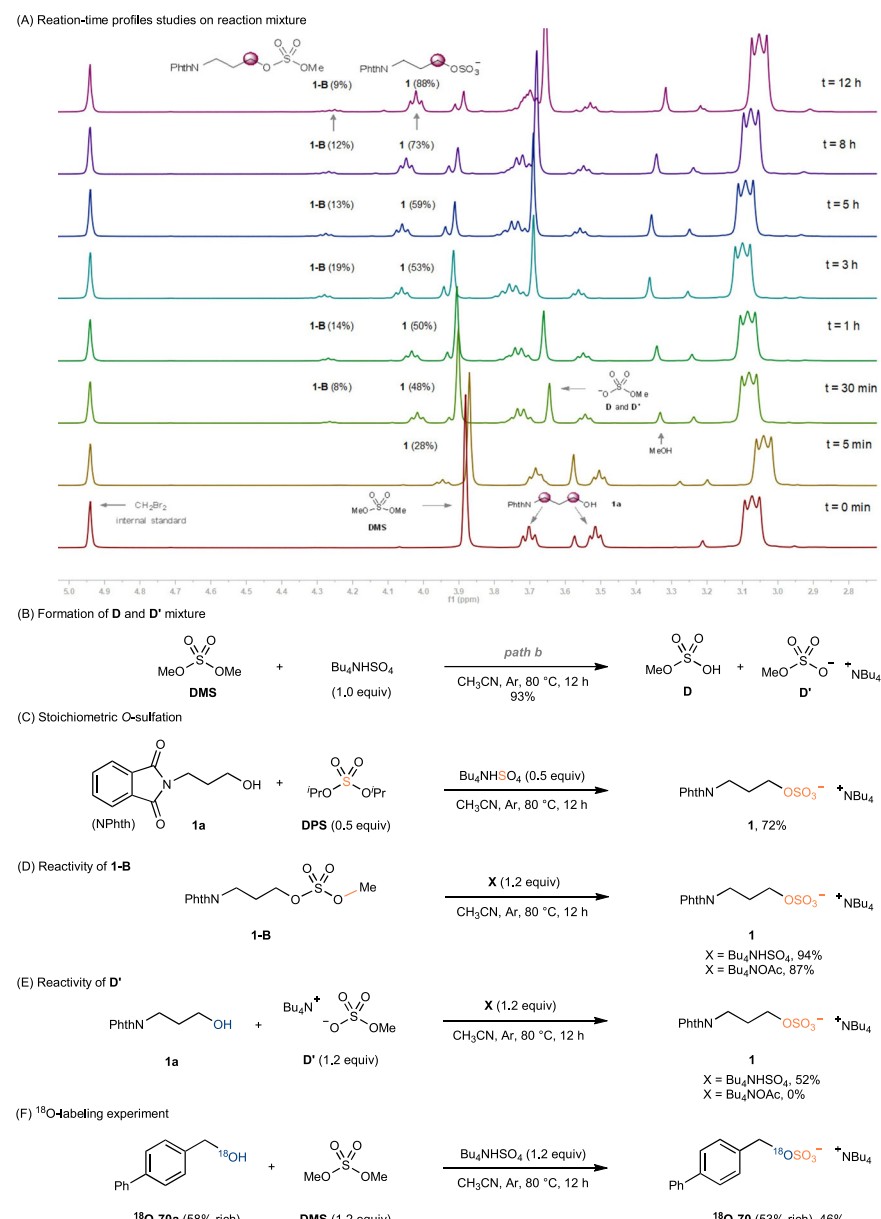

**Fig. 4 | Mechanistic experiments. A** Reaction-time profiles studies on **1** with DMS (1.2 equiv) in presence of Bu₄NHSO₄ (1.2 equiv) monitored by ¹H NMR spectra. **B** Formation of **D** and **D′** mixture. **C** Stoichiometric *O*-sulfation. **D** Reactivity of **1-B**. **E** Reactivity of **D′**. **F** ¹⁸O-labeling experiment.

stoichiometric amount of any individual sulfur source, suggests that the sulfate group in product originates from both of DMS/DPS, and Bu₄NHSO₄ (Fig. 4C). Treatment of methyl monosulfates (**1-B**) with Bu₄NHSO₄ or Bu₄NOAc resulted in the formation of the sulfated product **1** with high yields, indicating both bisulfate (HSO₄⁻) and acetate (AcO⁻) are effective nucleophilile to cleave the methyl protecting group in monosulfates **1-B** (Fig. 4D)[46]. On the contrary, the reactivity of methyl sulfate **D′** with **1a** differed significantly between tetrabutylammonium bisulfate and tetrabutylammonium acetate (Fig. 4E). The proton in tetrabutylammonium bisulfate was found to activate the sulfate group of methyl sulfate **D′**, enhancing the electrophilicity of sulfur atoms, as well as facilitating the leaving of the methyl group. The transformation of methyl sulfate **D′** to the final product is crucial for the easier purification of our reaction, as it eliminates potential contamination of the desired sulfate product with its similar polarity. To further elucidate the reaction pathway and the source of oxygen in the final product, we conducted ¹⁸O-labeling experiment with compound **70a**, which resulted in the

successful isolation of alcohol ¹⁸O-**70** in 46% yield with a nearly identical level of ¹⁸O enrichment (Fig. 4F). Given the formation of *S*–SO₃ product **20** rather than *O*–SO₃, the intact C–S bond demonstrated that the sulfation of R–OH proceeds via the formation of *O*–SO₃ bond rather than *C*–OSO₃, as the C–S bond is much weaker than C–O. Most importantly, the stereoselective retention of various chiral substrates (**46-59, 67-69**) in Fig. 3 clearly verifies the formation of *O*–SO₃ bond.

## Discussion
In summary, the discovery of an alternative activation method for dimethyl sulfate and diisopropyl sulfate allows for the versatile synthesis of organic sulfates. This work represents a notable advancement in the utilization of traditional reagents for significant chemical transformations. The mild reaction conditions and broad functional group tolerance make this method a powerful addition to the toolbox for late-stage sulfation of complex bioactive molecules. We anticipate that the generality of this *O*-sulfation and the ready

availability of the materials used in the transformation will broaden the application of organic sulfates in the comprehension of biological function and drug discovery.

## Methods

### General procedure for *O*-sulfation

To a 4.0 mL borosilicate vial equipped with a stir bar was added hydroxyl substrate (0.2 mmol, 1.0 equiv), dimethyl sulfate or diisopropyl sulfate or sodium pyrosulfat (0.24 mmol, 1.2 equiv) and tetrabutylammonium bisulfate (81.5 mg, 0.24 mmol, 1.2 equiv). The vial was evacuated and backfilled with argon for three times, then CH$_3$CN (1.0 mL, C = 0.2 M) was added. After stirring for 12 h at 80 °C, dibromomethane (14.0 μL, 0.2 mmol, 1.0 equiv) was added as an internal standard. The reaction mixture was diluted with CDCl$_3$, and the yield of tetrabutylammonium sulfates was determined by $^1$H NMR integration relative to the internal standard (standard: δ 4.94 ppm). To isolate the products, the concentrated reaction mixture was purified by flash chromatography on silica gel or plate chromatography on silica gel, to afford the product tetrabutylammonium sulfate salts.

## Data availability

The data reported in this paper are available within the article and its Supplementary Information files. Crystallographic data for the structures reported in this Article have been deposited at the Cambridge Crystallographic Data Centre, under deposition numbers CCDC2290725 (**8**). Copies of the data can be obtained free of charge via https://www.ccdc.cam.ac.uk/structures/. All data are available from the corresponding author upon request.

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

## Acknowledgements
This work was supported by the National Natural Science Foundation of China (Grant NO. 22001067 (J.L.), 22101081 (S.C.) and 22177125 (P.X.)), the Natural Science Foundation of Hunan Province (Grant NO. 2021JJ20022 (J.L.)) and the Youth Innovation Promotion Association of CAS (Grant NO.2020258 (P.X.)).

## Author contributions
J.L. and S.Y. conceived and designed the experiments. S.Y., G.D. and Y.Z. performed the experiments. J.L., C.S., P.X., B.Y. and S.Y. analyzed the results. J.L. wrote the paper and directed the project.

## Competing interests
J.L., S.Y., P.X. and G.D. are inventors on a patent application (number 202311012144.4, China), dealing with the use of dimethyl sulfate (DMS) and diisopropyl sulfate (DPS) for sulfation. The remaining authors declare no competing interests.
