## [Peer Review File · Nature Communications]

Dimethyl sulfate and diisopropyl sulfate as practical and versatile O-sulfation reagentsREVIEWER COMMENTS

Reviewer #1 (Remarks to the Author):

Yue and colleagues discuss the use of dimethyl and diisopropyl sulfate as a new sulfation reagent. The work is generally well-performed and sufficient in scope and novelty to warrant publication. However, I have some concerns with the paper:

- 1) What advantage does the reagent have over other best in class sulfation agents? Already known to some extent eg amine-SO₃ reagents.
- 2) The reagent is known to be horribly toxic and mutagenic- this needs to be dealt with in the paper and SI.
- 3) Has the use of the CF₃ derivative of the reagent been considered (less toxic)
- 4) If the bisulfate acts as protic catalysis why is it so special over a simple protonation method?
- 5) Has N or S sulfation been tested to show generality?
- 6) In Table 1 what is the solubility profile of KHSO₄ in MeCN - could this be a reason?
- 7) Have other alkyl sulfates been tested as comparitors to Bu₄NHSO₄?
- 8) The work of Kowalska 2012 Kowalska, J., Osowniak, A., Zuberek, J. & Jemielity, J. Synthesis of nucleoside phosphosulfates. *Bioorg. Med. Chem. Lett.* 22, 3661–3664 (2012). Should be considered as has a bearing on the work proposed
- 9) Why is reaction not at bp of MeCN - 80 not 82C?
- 10) Do the arylsulfates undergo rearrangement to the phenol and C-sulfonates?
- 11) Why is the selection of DPS over DMS justified - unclear to me why one used over another?
- 12) Why is the need for Na₂S₂O₇ required in some examples - mechanistic purpose?
- 13) With more hindered alcohols why does the more hindered DPS work better than DMS?
- 14) Any selectivity between primary/secondary/tertiary alcohols? (not phenol)
- 15) Would the ¹⁸O experiment be better on the DMS? Eg could the sulfate not come from Bu₄NHSO₄ a bit like that Bu₃NSO₃ reagent from a few years ago?
- 16) Didn't follow this sentence "outcome clearly demonstrated that the sulfation proceeds via the formation of O-SO₃ bond rather than C-OSO₃, as evidenced by the" was unclear as the level of ¹⁸O incorporation is not specified or data on isotopic MS in place?
- 17) Recent work by Lara Malins and others to be cited on sulfating strategies is lacking.

Supporting Information:

- 1) Why does 11, 19, 58, 65 give low conversion/unpurifiable? Needs to be clarified in text too.
- 2) ¹⁸O experiment the quench should be incorporated not ¹⁶OH₂
- 3) ¹⁸O % incorporation not given - essential.
- 4) ¹⁸O MS data eg screenshots vs abundance predictions needed.
- 5) Need a safety statement re DMS and DPS usage.
- 6) ¹⁹F NMR for 36 has 2 peaks?
- 7) NMR for 58 has a drift on baseline why?
- 8) Which compounds are NOVEL?

Reviewer #2 (Remarks to the Author):

The submitted paper deals with a pioneering method for O-sulfation, a crucial post-translational modification and biotransformation within the metabolism of bioactive molecules. The introduction begins by emphasizing the significance of O-sulfation in various biomolecules and highlights the challenges in their synthesis. Traditional methods, such as sulfur trioxide-nitrogen base complexes, face limitations, prompting the need for alternative strategies like early-stage sulfation. The authors introduce dialkyl sulfates, specifically dimethyl sulfate (DMS) and diisopropyl sulfate (DPS), as promising sulfate diesters for O-sulfation.

The proposed method involves activating DMS or DPS under tetrabutylammonium bisulfate, enhancing their reactivity for efficient sulfate transfer. Extensive optimization studies identified DMS as the optimal sulfate source, coupled with Bu₄NHSO₄ as the ideal activation agent. Control experiments validated the necessity of both DMS and Bu₄NHSO₄ for successful sulfation. The method demonstrates versatility, effectively sulfating a range of primary and secondary alcohols with DMS and DPS, respectively.

The authors showcase the broad applicability of their method to various compounds, including primary alcohols, secondary alcohols, amides, halides, nitrates, ethers, alkenes, alkynes, boronic esters, aldehydes, sulfones, esters, ketones, sulfonamides, aromatics, and heterocycles. Additionally, the method proves effective for the gram-scale synthesis of sulfated products.

The paper extends the method's application to carbohydrates, amino acids, and steroids. Various sugar derivatives, amino acids, and complex natural products were successfully sulfated under mild conditions, showcasing the method's practicality for late-stage modifications of bioactive molecules.

Results from control experiments provide insights into the reaction mechanism, indicating the involvement of key intermediates such as methyl monosulfates and methyl sulfate D'. Notably, the review includes ¹⁸O-labeling experiments, confirming the formation of O-SO₃ bonds in the sulfation process. Of note, the manuscript is accompanied by extensive supplementary information (122 pages)

In conclusion, the paper presents the discovery of a novel activation method for dimethyl sulfate and diisopropyl sulfate as a significant advancement in the synthesis of organic sulfates. The method's mild reaction conditions, broad functional group tolerance, and applicability to complex biomolecules position it as a powerful tool for late-stage sulfation and drug discovery. The authors anticipate the widespread use of this O-sulfation method in comprehending biological functions and advancing drug discovery.

There are no major problems in the manuscript, just several minor comments can be found in the annotated manuscript and supplementary file.

Reviewer #3 (Remarks to the Author):

The manuscript by Yue et al. represents a noteworthy progression in the field of O-sulfation chemistry. Notably, the authors have elucidated a novel reactivity exhibited by dimethyl sulfate and diisopropyl sulfate, serving as innovative O-sulfation reagents under acidic conditions. The transformations outlined predominantly involve the O-sulfation of alcohols,

phenols, and carbohydrates, resulting in the production of sulfonated chemicals. This research is particularly intriguing as it refines the reactivity of dimethyl sulfate and diisopropyl sulfate, traditionally acknowledged for their robust alkylating properties, by harnessing their potential as sulfation reagents.

The authors demonstrate the method work with a broad array of electronically and sterically diverse alcohols, carbohydrates, amino acids, and natural products, especially for showcasing selectivity on substrates (55-58). It is noteworthy, however, that the method exhibits limitations in the sulfation of tertiary alcohols and phenols when employing DMS or DPS as the reagents. The authors, in an attempt to address the sulfation of phenols, utilized sodium pyrosulfate as a reagent, a tangent that is not directly pertinent to the focus of this manuscript. Nevertheless, this limitation does not diminish the significance of the work, as the manuscript furnishes compelling evidence that dimethyl sulfate and diisopropyl sulfate can indeed serve as O-sulfation reagents for alcohols and carbohydrates, exhibiting a novel reactivity.

The elucidation of how the reactivity of dimethyl sulfate was modulated presents an intriguing aspect of this study. While the paper does furnish some evidence for the proposed pathway, there remain substantial mechanistic inquiries that warrant attention, particularly concerning the substantiation of the reactive intermediate. Several mechanistic hypotheses pivot on the existence of methyl monosulfates (1-B). As it currently stands, additional experiments are imperative to offer more compelling evidence regarding its presence. Given the substantial emphasis on mechanistic elucidation within this manuscript, these unresolved issues pose impediments to recommending it for publication.

In summary, this work presented by Li and their colleagues represents an important contribution to the field, showcasing a significant advancement in O-sulfation chemistry. The broad substrate scope and excellent selectivity observed in this study open up new avenues for the synthesis of complex and valuable molecules. The compounds are appropriately characterized, and manuscript was well-written. In my opinion, the manuscript merits acceptance for publication in Nature Communications, pending the authors' careful attention to the following suggestions raised.

Conditions optimization:

1. Given the author's emphasis on the importance of acidic conditions in this reaction, have the authors attempted to incorporate the corresponding acid for entries 5-6 in Table 1?
2. On the other hand, the yield with a counter anion other than HSO₄ is quite good for entries 4-6, especially for entry 5, even in the absence of acid. Does the author have an explanation for how the product was formed in this case?

Substrates scope:

3. All the products were well-characterized; however, only NMR yield was provided for substrates (11, 19-24, 58, 61, 65, 66).
4. For conditions d in Table 2, several other counter cations like Na⁺, K⁺, were used. How did the authors determine that these were not included in the final products?

Mechanistic studies:

5. In Figure 2 A, the ¹H NMR spectrum of methyl monosulfate (1-B) does not appear consistent with it being the main intermediate, especially given its low integration. Without integrations of the key signals, it is hard to discern the correlation between the new peaks in the ¹H NMR and that they correlate to the main intermediate. Integration of these peaks in

relation to the standard in ^1H spectra would be helpful.

6. According to the proposed mechanism in Figure 1C-ii, Bu_4NHSO_4 would react with intermediate B. Why wouldn't it react with DMS at the beginning, given that DMS is less sterically hindered than intermediate B?

7. Have the authors investigated the formation of SO_3 during the reaction? They could attempt to detect and capture SO_3 by mixing the DMS and Bu_4NHSO_4 . This would enhance the understanding of the reaction mechanism.

8. Determining the source of the 'sulfur' in the product is crucial, and conducting experiments with ^{34}S labeled Bu_4NHSO_4 and/or DMS would provide valuable insights.

9. In drawing the mechanism (Figure 1C-ii, from A to B), based on the arrows the authors have depicted, it suggests the formation of a methoxide anion. However, considering the acidic conditions, the existence of a methoxide anion is improbable. The arrow might be better represented as originating from the oxygen of the alcohol rather than the hydrogen. SI and others:

10. It appears that the R_f values for most products are identical. Could the authors please confirm this observation? Additionally, it's noted that most of the products share the same physical state, described as a light yellow oil.

11. The abbreviation for diisopropyl sulfate is not consistent between the main text and the SI. Please ensure uniformity in the usage of abbreviations throughout the manuscript.

12. Some typos: Figure 1C-li -> Figure 1C-ii; line 93: 12 h -> 12 hours; lines 95-98: entry -> entries; line 97: slower yield -> lower yield...

Referees' comments:

Reviewer 1 (Remarks to the Author):

We would like to thank the reviewer for their time and consideration of this manuscript. We are grateful for their advice/corrections to improve the quality of our initial submission. Please find below, a list of comments, responses, and the changes made to the manuscript and/or supplementary information.

Yue and colleagues discuss the use of dimethyl and diisopropyl sulfate as a new sulfation reagent. The work is generally well-performed and sufficient in scope and novelty to warrant publication.

We are grateful to the reviewer for this comment.

However, I have some concerns with the paper:

1) What advantage does the reagent have over other best in class sulfation agents? Already known to some extent eg amine-SO₃ reagents.

We thank the reviewer to pointing this out. To best of our knowledge, other classic sulfation agents include sulfonyl chloride, SO₃-amide (DMF) complex and sulfonyl imidazolium salts. With their increased stability, readily availability and straight maneuverability, DMS and DPS have shown a broad substrate scope and wide functional group tolerance, spanning from carbohydrates, amino acids to natural products.

2) The reagent is known to be horribly toxic and mutagenic- this needs to be dealt with in the paper and SI.

We appreciate the reviewer's concern for hazard statements of DMS and DPS.

The safety warning for DMS and DPS has been added in the revised manuscript at Table 2's caption:

"Note: The liquid DMS and DPS are volatile and toxic. Exercise extreme caution when handling the liquid."

Supporting Information (page 5):

"Note: Alkyl sulfate is highly toxic, so the reaction process should be handled with care. It is recommended to work in a fume hood."

3) Has the use of the CF₃ derivative of the reagent been considered (less toxic)

We appreciate the reviewer's suggestion to explore DMS fluoride derivatives as potential sulfation reagents, as they could potentially offer a safer and more efficient alternative. However, it is important to note that the synthesis of bis(trifluoromethyl) sulfate needs the use of fluorine gas, which is an exceedingly complex process, with only 0.1% yield (*Journal of the Chemical Society, Perkin Transactions 1*, **1979**, 2675-2678). Even if practical synthesis methods were to be developed, it is reasonable to doubt whether dimethyl sulfate alone can effectively carry out the sulfation process. The reaction mechanism from dimethyl sulfate suggests that the released HO-CF₃ is highly unstable and prone to decomposition into fluorophosgene **III** and fluoride. As a result, alcohol substrate would likely react with fluorophosgene **III** to form the corresponding fluoroformate **IV**, which would then be susceptible to attack from any nucleophile present in the reaction mixture, ultimately leading to sulfation failure. Furthermore, our attempts to synthesize DPS fluoride derivatives were not successful under the various conditions examined.

Conversely, diaryl sulfates can be easily obtained; however, they exhibit a lack of reactivity due to their high stability.

These results are included in the SI (page 6):

Entry	Variation from the standard conditions	Yield of 1 (%) ^a
1		84%
2		74%
3		87%
4		57%
5		n.d.
6		n.d.
7		n.d.

^a Yields were determined by ¹H NMR using dibromomethane as an internal standard.

4) If the bisulfate acts as protic catalysis why is it so special over a simple protonation method?

We greatly appreciate the reviewer for highlighting the crucial role of bisulfate in the sulfation process. HSO₄⁻ plays two essential functions in sulfation:

1. Activation of DMS or DPS;
2. Nucleophilic cleavage of the intermediate **1-B**.

After activation of dialkyl sulfations by protic acids with other anions (HX), the nucleophilic X⁻ may also cleave the intermediate methyl monosulfates **1-B**, and generate the product **C** as well as the methyl derivatives **D-1**. Among these compounds, only **D** (X = HSO₄⁻) can proceed with sulfation further. In contrast, sulfation will be halted at this step without the presence of HSO₄⁻. Therefore, the special role of bisulfate lies in acting as both a proton acid activator and a source of sulfate.

5) Has N or S sulfation been tested to show generality?

We agree that it would be appropriate to further extend the substrate scope. In response to this feedback, we have tested six substrates, two of which are thiol and thiophenol, affording *S*-sulfated products. Unfortunately, our method is not suitable to *N*-sulfation. This is because the protonation of N leads to the formation of ammonium or iminium salts, which cannot undergo nucleophilic attack on DMS or DPS.

S-sulfation:

^aReaction conditions as shown in table 1

^bNa₂S₂O₇ (2.0 equiv) and Bu₄NHSO₄ (1.1 equiv) were used.

N-sulfation:

all failed with no conversion

In the revised manuscript:

“This new protocol was also successful in producing *S*-sulfation products 20 and 45 for thiol and thiophenol. However, nitrogen compounds such as amine and imine (Supplementary Table 4) did not yield any desired sulfation products, and all starting materials were recovered.”

The Table 2 was updated as below, and new data can be found in revised SI (page 17, 30).

Table 2. Substrate scope of *O,S*-sulfation^{a,b}

R-OH/SH		DMS or DPS (1.2 equiv) Bu ₄ NHSO ₄ (1.2 equiv) CH ₂ Cl ₂ , Ar, 80 °C, 12 h		R-O'S-SO ₃ ⁻ NBu ₄		MeO-S(=O)-OMe DMS		PrO-S(=O)-OPr DPS	
Primary alcohol/thiol									
1, 80% ^a (10 mmol, 4.2 g, 79%)	2, 67% ^a	3, R = F, 71% ^a	4, R = NO ₂ , 80% ^a	5, R = CN, 72% ^a	6, R = t -Bu, 81% ^a	7, R = OMe, 92% ^a	8, 80% ^a		
9, 81% ^a	10, 90% ^a	11, 70% ^b (92% ^c)	12, 87% ^b	13, 63% ^b	14, 55% ^b	15, 79% ^b	16, 70% ^b	17, 72% ^b	18, 80% ^a
19, 24% ^b (34% ^c)	20, 41% ^a	21, 84% ^b	22, 41% ^b	23, 73% ^b	24, 72% ^b	25, 75% ^b (94% ^c)	26, 84% ^b	27, 80% ^b	28, 21% ^b (33% ^c)
Secondary alcohol									
29, 29% ^b (40% ^c)	30, 85% ^d	31, 77% ^d	32, 75% ^d	33, 85% ^d	34, 88% ^d	35, R = Et, 82% ^d	36, R = t -Bu, 87% ^d	37, 90% ^d	38, 81% ^d
Tertiary alcohol									
39, 83% ^d	40, 75% ^d	41, 68% ^d	42, 80% ^d	43, 83% ^d	44, 82% ^d	45, 83% ^b	46, 82% ^d	47, 81% ^d	48, 83% ^d
(Thio)phenol									
49, 85% ^d	50, 88% ^d	51, 80% ^d	52, 80% ^d	53, 83% ^d	54, 82% ^d	55, 83% ^b	56, 82% ^d	57, 81% ^d	58, 83% ^d

6) In Table 1 what is the solubility profile of KHSO₄ in MeCN - could this be a reason?

We agree with the reviewer's observation that the poor solubility of KHSO₄ in MeCN is responsible for its failure in sulfation. Less than 1 mg (0.6 mg) of inorganic salt KHSO₄ could be dissolved soluble in MeCN (1 mL) at 80 °C: to a 4-mL borosilicate vial was added KHSO₄ (32.7 mg) and MeCN (1.0 mL). After heating to 80 °C, the hot mixture was filtered, and residue solid was then dried under vacuum, resulting in the recovery of KHSO₄ (32.1 mg).

Moreover, the combined use of Bu₄NBF₄ with KHSO₄ proved to be effective in this transformation due to the increased solubility of the activator (HSO₄⁻).

7) Have other alkyl sulfates been tested as comparitors to Bu₄NHSO₄?

We thank you the reviewer for this suggestion. We have attempted several tetra-alkyl ammonium bisulfates (from C1 to C6), all of which delivered the desired product in 70% to 84% yield. Although the reasons for the differences in yield between these alkyl substituents are unclear, we suspect that the solubility and stability of the corresponding organosulfate ammonium salt may play a role, as the ⁿBu can ensure product stability while increasing solubility (*Chem. Commun.* **2019**, 55, 4319-4322).

These results can be found in revised Supporting Information (page 6):

Entry	R ₄ NHSO ₄	Yield of 1 (%) ^a
1	Me ₄ NHSO ₄	76%
2	Et ₄ NHSO ₄	75%
3	ⁿ Pr ₄ NHSO ₄	74%
4	ⁿ Bu ₄ NHSO ₄	84%
5	ⁿ Hex ₄ NHSO ₄	70%

^a Yields were determined by ¹H NMR using dibromomethane as an internal standard.

8) The work of Kowalska 2012 Kowalska, J., Osowniak, A., Zuberek, J. & Jemielity, J. Synthesis of nucleoside phosphosulfates. *Bioorg. Med. Chem. Lett.* **22**, 3661–3664 (2012). Should be considered as has a bearing on the work proposed

We added this new reference as No. 46, which is about nucleophile of bis(tributylammonium) sulfate. Therefore, we reorganized the numbering of the reference.

In the revised manuscript:

“These outcome confirmed the crucial role of Bu₄NHSO₄ for enhanced solubility of sulfate product **1** as tetrabutylammonium salt as well as facile removal of the methyl group in sulfate monoester **B**.⁴⁶”

“Treatment of methyl monosulfates (**1-B**) with tetrabutylammonium bisulfate (Bu₄NHSO₄) or tetrabutylammonium acetate (Bu₄NOAc) resulted in the formation of the sulfated product **1** with high yields, indicating both bisulfate (HSO₄⁻) and acetate (AcO⁻) are effective nucleophilile to cleavage the methyl protecting group in monosulfates **1-B** (Figure 2B).⁴⁶”

9) Why is reaction not at bp of MeCN - 80 not 82 C?

There is no difference in our reaction between 80 or 82 °C. We specifically selected 80 °C based on safety considerations. By conducting the reaction slightly below the MeCN boiling point (82 °C), we can avoid reflux, which is both favorable for the reaction and minimizes any potential hazards.

10) Do the arylsulfates undergo rearrangement to the phenol and C-sulfonates?

We appreciate the reviewer's concern about the rearrangement. Under our reaction conditions, no rearrangement of arylsulfates occurred. As an example, we randomly selected substrate **38**, which only gives phenol arisen from the hydrolysis in 26% yield under our standard conditions.

11) Why is the selection of DPS over DMS justified - unclear to me why one used over another?

We thank the reviewer for pointing this out. We specifically chose to study DMS as a novel concept to demonstrate that DMS can function not only as a well-known methylating reagent, but also as a sulfating reagent. For practical purposes, DPS exhibits higher reactivity than DMS for most substrates, including primary and secondary alcohols. Moreover, it is less toxic with its larger molecule weight and higher boiling point.

12) Why is the need for Na₂S₂O₇ required in some examples - mechanistic purpose?

We regret that we were not able to articulate clearly in our original submission. As we originally mentioned, phenols, tertiary alcohols, and N–OH compounds are generally considered to have weaker nucleophilicity compared to primary and secondary alcohols. This makes them less reactive and sluggish towards attacking dialkyl sulfates, even with acid activation. On the other hand, Na₂S₂O₇, on the other hand, possesses a stronger electrophilic sulfur atom (as a form of “anhydride” of sodium sulfate) and a better leaving group (SO₄²⁻). These unique properties makes it more favorable for the sulfation to occur, especially when dealing with challenging substrates.

The mechanistic purpose was added in revised SI (page 7):

13) With more hindered alcohols why does the more hindered DPS work better than DMS?

We thank the reviewer for pointing this out. Based on the proposed mechanism in Figure 1C-ii, there may be two factors leading to the different reactivity between DMS and DPS:

1. The steric hindrance between **A** and substrate ROH.

2. The leaving group (${}^i\text{PrO}^-$ vs MeO^-) at step **A**→**B**.

3. Stability of carboncation (${}^i\text{Pr}^+$ vs Me^+) at the step **B**→**C**;

We agree with the reviewer that the larger group of ${}^i\text{Pr}$ in DPS is not benefit for the nucleophilic attack of hindered alcohols. However, the better leaving group of ${}^i\text{PrO}$ (pK_a : ${}^i\text{PrOH}$ 29.0, MeOH 30.3; *JOC*, **1980**, 45, 3295-3299) and the greater stability of ${}^i\text{Pr}$ are facile to the sulfation. The reaction may be primarily influenced by the latter two factors.

14) Any selectivity between primary/secondary/tertiary alcohols? (not phenol)

We thank the reviewer for additional discussion on the selective sulfation among primary, secondary, tertiary alcohols.

We agree with the reviewer for more discussion on the selective sulfation. As demonstrated in the manuscript, we were able to achieve selectivity between primary and secondary alcohols (**57-59**), and found that steric effects play a significant role in determining the selectivity. Now, we further investigated the selectivity between primary and tertiary alcohols (**60**). However, we acknowledge that achieving selectivity among substrates with similar steric hindrance has been more challenging. In the case of compound **71s** with two secondary hydroxyl groups, both can be sulfated. We would like to thank the reviewer for bringing attention to these points and we will continue our efforts to advance selectivity in sulfation reactions.

a). Selective sulfation between primary and tertiary hydroxyl group

b). Sulfation between among secondary hydroxyl groups

In the revised manuscript, we add more discussion on the selectivity:

“The selectivity in these reactions appears to be mainly controlled by steric effects, as no distinction was observed among substrates with similar hindrance (see Supplementary Information).”

The new data for sulfation on substrate 71s, are included in the revised supplementary information (page 40).

15) Would the ${}^{18}\text{O}$ experiment be better on the DMS? Eg could the sulfate not come from Bu_4NHSO_4 a bit like that Bu_3NSO_3 reagent from a few years ago?

We agree with the reviewer that the ${}^{18}\text{O}$ -DMS is a direct method to determine the source of sulfate. However, there are two types of ${}^{18}\text{O}$ -labeled DMS depending on oxygen location: ${}^{18}\text{O}$ -DMS **1** and ${}^{18}\text{O}$ -DMS **2**.

The synthesis of type **1** from Me^{18}OH may be not difficult, but it will be cleaved by substrate ROH and exchanged with Bu_4NHSO_4 during the reaction, providing no hints for our mechanism investigation. However, the synthetic route to **2** is unknown, regardless of whether it is from ${}^{18}\text{O}_2$ or $\text{H}_2{}^{18}\text{O}$. We are unsure if we can complete this complicated and time-consuming procedure.

On the contrary, we have more evidence to support the source of sulfate. Please see our reply to comment 8 of reviewer 3 for more details.

As for the potential formation of $\text{Bu}_3\text{N}\cdot\text{SO}_3$ complex (*Chem. Commun.* **2019**, 55, 4319-4322), treatment of DMS and Bu_4NHSO_4 (1.0 equiv) at 80 °C, affords methyl sulfates (MeOSO_3^-) in 93% yield, with no detectable $\text{Bu}_3\text{N}\cdot\text{SO}_3$ by ^1H NMR spectra and LC-MS, nor for SO_3 by GC-MS.

^1H NMR of $\text{Bu}_3\text{N}\cdot\text{SO}_3$: (400 MHz, CDCl_3 , ppm) δ 3.32 – 3.22 (m, 6H, $\text{N}(\text{CH}_2)_3$), 1.87 – 1.73 (m, 6H, CH_2), 1.37 (dt, $J = 7.4$ Hz, 6H, CH_2), 0.98 (t, $J = 7.4$ Hz, 9H, CH_3).

LC-MS of the above mixture: 110.9757 was HRMS of the methyl sulfate (MeOSO_3^-), no information was found for $\text{Bu}_3\text{N}\cdot\text{SO}_3$ (exact mass: 265.1712).

GC-MS of the above mixture: no mass signal was found for SO_3 ($m/z = 79.9568$).

In the revised manuscript, we add:

“DMS reacts with tetrabutylammonium bisulfate (Bu_4NHSO_4) to yield a mixture of **D** and **D'** (Figure 2B) rather than SO_3 complex, which substantiates the possible path b in the reaction (Figure 1C-ii).”

16) Didn't follow this sentence "outcome clearly demonstrated that the sulfation proceeds via the formation of O–SO₃ bond rather than C–OSO₃, as evidenced by the" was unclear as the level of O¹⁸ incorporation is not specified or data on isotopic MS in place?

Please see our reply to the reviewer's following comment 3 on Supporting Information.

17) Recent work by Lara Malins and others to be cited on sulfating strategies is lacking.

We regret that we did not include this literature in our original submission. We have added the suggested references (No. 30) to the second paragraph of the introduction:

"Sulfur trioxide-nitrogen base complexes are the most commonly used reagents for sulfating various molecular motifs containing alcoholic, phenolic, amino, thiol and other functional groups.²⁷⁻³⁰"

Supporting Information:

1) Why does 11, 19, 58, 65 give low conversion/unpurifiable? Needs to be clarified in text too.

We thank the reviewer for pointing this out. These four substrates can all be purified, allowing us to include NMR and HRMS data in the original supplementary information. We preferred their yields determined by ¹H NMR over the isolated yields because of the significant differences between them, which were caused by their decomposition on chromatography.

As a response to similar comments raised by reviewer 3, both the purified compound yields and the yields determined by NMR are reflected in the new table, and the new data are included in the revised supplementary information.

Compound **11**: revised yield is 70%^b (92%^c), previously 92%^{b,c}.

Compound **19**: revised yield is 24%^b (34%^c), previously 34%^{b,c}.

Compound **24** (renumbered **25**): revised yield is 75%^b (94%^c), previously 94%^b.

Compound **27** (renumbered **28**): revised yield is 21%^b (33%^c), previously 33%^{b,c}.

Compound **28** (renumbered **29**): revised yield is 29%^b (40%^c), previously 40%^{b,c}.

Compound **58** (renumbered **61**): revised yield is 70%^c (88%^f), previously 88%^{e,f}.

Compound **61** (renumbered **64**): revised yield is 45%^a (60%^f), previously 60%^{a,f}.

Compound **65** (renumbered **68**): revised yield is 71%^a (90%^f), previously 90%^{a,f}.

Compound **66** (renumbered **69**): revised yield is 38%^c (49%^f), previously 49%^{e,f}.

In the revised manuscript, we add:

"Due to the lability of organic sulfates on chromatography, such as **11**, **19** and **25**, the purified compound yields was significantly lower than the yields determined by ¹H NMR."

2) ¹⁸O experiment the quench should be incorporated not 16OH₂

We thank the reviewer for pointing this out. To prevent contamination from ¹⁶O, we have re-run the reaction without the quenching step.

We have added this new procedure in the revised SI (page 49):

"After stirring for 12 h at 80 °C, the reaction was cooled down to room temperature, and concentrated by rotary evaporation."

3) ¹⁸O % incorporation not given - essential.

We agree with the reviewer's assertion that the incorporation of ¹⁸O in final is essential for reaction mechanism. As a result, we have made significant updates to ensure that both the starting material and the resulting product possess a nearly identical enrichment of ¹⁸O. The data presented in our study demonstrates that no erosion of ¹⁸O occurs, thereby supporting the notion of an intact C–O bond throughout the entire reaction process.

Another solid evidence is that various chiral substrates (**46-59**, **67-69**) proceed with stereoselective retention. Most importantly, R-SH substrates affords S-SO₃ products **20** and **45** without formation of O-SO₃. The intact C-S bond clearly demonstrated that the sulfation of R-OH proceeds via the formation of O-SO₃ bond rather than C-OSO₃, as the C-S bond is much weaker than C-O.

We have indicated this result in the revised manuscript (Figure 2F):

“To further elucidate the reaction pathway and the source of oxygen in the final product, we conducted ¹⁸O-labeling experiment with compound **70a**, which resulted in the successful isolation of alcohol ¹⁸O-**70** in 46% yield with a nearly identical level of ¹⁸O enrichment.”

“Given the formation of S-SO₃ products **20** rather than O-SO₃, the intact C-S bond demonstrated that the sulfation of R-OH proceeds via the formation of O-SO₃ bond rather than C-OSO₃, as the C-S bond is much weaker than C-O. Most importantly, the stereoselective retention of various chiral substrates (**46-59**, **67-69**) in Table 3 clearly verifies the formation of O-SO₃ bond.”

The new data, and ¹⁸O MS spectra is added in the revised supplementary information (page 50, 57):

4) ¹⁸O MS data eg screenshots vs abundance predictions needed.

Addressed. See our reply to the above comment 3.

5) Need a safety statement re DMS and DPS usage.

Addressed. Please see our reply to the very beginning comment 2.

6) ¹⁹F NMR for **36** has 2 peaks?

We thank the reviewer for pointing this out. The presence of a small impure substrate **36** (updated No. **37**) led to 2 signal peaks in ^{19}F -NMR. We have re-run the reaction for compounds **37**. The yields of the purified compounds are reflected in the new Table 2, and the new NMR spectra are included in the revised supplementary information (page 98).

Compound **37**: revised yield is 90% (previously 91% yield).

7) NMR for 58 has a drift on baseline why?

The drift on baseline is probably caused by shimming from the instrument, or the structure of **58** containing a phenol group. After Na^+ exchange, the sodium sulfate **58** (updated No. **61**) with longer acquisition times and more delay time to relax between pulses can quiet down the baseline and allow for better phasing. The new NMR spectra is included in the revised supplementary information (page 122).

8) Which compounds are NOVEL?

In the SI, all the new compounds have been thoroughly characterized by ^1H NMR, ^{13}C NMR and HRMS. Additionally, the known compounds (**24**, **40** and **42**) have been characterized alongside appropriate references. Regarding the reviewer's comment about "NEVEL", it might refer to the novelty of the structure, indicating that it has not been previously reported. In our substrate table, the *N*-OSO₃ (**31**, **32**) and tertiary sulfate (**30**) compounds are novel molecules that are not otherwise readily accessible.

Reviewer 2 (Remarks to the Author):

We appreciate the reviewer's meticulous analysis of our work, and their advice/corrections to improve the quality of our initial submission. We agree with all comments made by this reviewer and present here the changes we made to the revised submission.

The submitted paper deals with a pioneering method for O-sulfation, a crucial post-translational modification and biotransformation within the metabolism of bioactive molecules. The introduction begins by emphasizing the significance of O-sulfation in various biomolecules and highlights the challenges in their synthesis. Traditional methods, such as sulfur trioxide-nitrogen base complexes, face limitations, prompting the need for alternative strategies like early-stage sulfation. The authors introduce dialkyl sulfates, specifically dimethyl sulfate (DMS) and diisopropyl sulfate (DPS), as promising sulfate diesters for O-sulfation.

The proposed method involves activating DMS or DPS under tetrabutylammonium bisulfate, enhancing their reactivity for efficient sulfate transfer. Extensive optimization studies identified DMS as the optimal sulfate source, coupled with Bu₄NHSO₄ as the ideal activation agent. Control experiments validated the necessity of both DMS and Bu₄NHSO₄ for successful sulfation. The method demonstrates versatility, effectively sulfating a range of primary and secondary alcohols with DMS and DPS, respectively.

The authors showcase the broad applicability of their method to various compounds, including primary alcohols, secondary alcohols, amides, halides, nitrates, ethers, alkenes, alkynes, boronic esters, aldehydes, sulfones, esters, ketones, sulfonamides, aromatics, and heterocycles. Additionally, the method proves effective for the gram-scale synthesis of sulfated products.

The paper extends the method's application to carbohydrates, amino acids, and steroids. Various sugar derivatives, amino acids, and complex natural products were successfully sulfated under mild conditions, showcasing the method's practicality for late-stage modifications of bioactive molecules.

Results from control experiments provide insights into the reaction mechanism, indicating the involvement of key intermediates such as methyl monosulfates and methyl sulfate D'. Notably, the review includes ^{18}O -labeling experiments, confirming the formation of O-SO₃ bonds in the sulfation process. Of note, the manuscript is accompanied by extensive supplementary information (122 pages)

In conclusion, the paper presents the discovery of a novel activation method for dimethyl sulfate and

diisopropyl sulfate as a significant advancement in the synthesis of organic sulfates. The method's mild reaction conditions, broad functional group tolerance, and applicability to complex biomolecules position it as a powerful tool for late-stage sulfation and drug discovery. The authors anticipate the widespread use of this O-sulfation method in comprehending biological functions and advancing drug discovery.

We thank the reviewer for their careful appraisal of our work.

There are no major problems in the manuscript, just several minor comments can be found in the annotated manuscript and supplementary file.

We are grateful to the reviewer for their corrections on our typographical errors, and all the suggestions for improving our manuscript.

Reviewer 3 (Remarks to the Author):

We thank the reviewer for the time taken to consider our manuscript. We agree with all comments made by this reviewer and present here the changes we made to the revised submission.

The manuscript by Yue et al. represents a noteworthy progression in the field of *O*-sulfation chemistry. Notably, the authors have elucidated a novel reactivity exhibited by dimethyl sulfate and diisopropyl sulfate, serving as innovative *O*-sulfation reagents under acidic conditions. The transformations outlined predominantly involve the *O*-sulfation of alcohols, phenols, and carbohydrates, resulting in the production of sulfonated chemicals. This research is particularly intriguing as it refines the reactivity of dimethyl sulfate and diisopropyl sulfate, traditionally acknowledged for their robust alkylating properties, by harnessing their potential as sulfation reagents.

The authors demonstrate the method work with a broad array of electronically and sterically diverse alcohols, carbohydrates, amino acids, and natural products, especially for showcasing selectivity on substrates (55-58). It is noteworthy, however, that the method exhibits limitations in the sulfation of tertiary alcohols and phenols when employing DMS or DPS as the reagents. The authors, in an attempt to address the sulfation of phenols, utilized sodium pyrosulfate as a reagent, a tangent that is not directly pertinent to the focus of this manuscript. Nevertheless, this limitation does not diminish the significance of the work, as the manuscript furnishes compelling evidence that dimethyl sulfate and diisopropyl sulfate can indeed serve as *O*-sulfation reagents for alcohols and carbohydrates, exhibiting a novel reactivity.

We thank the reviewer for their careful appraisal of our work.

The elucidation of how the reactivity of dimethyl sulfate was modulated presents an intriguing aspect of this study. While the paper does furnish some evidence for the proposed pathway, there remain substantial mechanistic inquiries that warrant attention, particularly concerning the substantiation of the reactive intermediate. Several mechanistic hypotheses pivot on the existence of methyl monosulfates (1-B). As it currently stands, additional experiments are imperative to offer more compelling evidence regarding its presence. Given the substantial emphasis on mechanistic elucidation within this manuscript, these unresolved issues pose impediments to recommending it for publication.

We agree with the reviewer's concern on this point. Please see our reply to the comments 5-9 on the mechanistic elucidation.

In summary, this work presented by Li and their colleagues represents an important contribution to the field, showcasing a significant advancement in *O*-sulfation chemistry. The broad substrate scope and excellent selectivity observed in this study open up new avenues for the synthesis of complex and valuable molecules. The compounds are appropriately characterized, and manuscript was well-written. In my opinion, the

manuscript merits acceptance for publication in Nature Communications, pending the authors' careful attention to the following suggestions raised.

We thank the reviewer for this comment.

Conditions optimization:

1. Given the author's emphasis on the importance of acidic conditions in this reaction, have the authors attempted to incorporate the corresponding acid for entries 5-6 in Table 1?

We thank the reviewer for this suggestion. The combined use of Bu_4NBF_4 and KHSO_4 was observed to be more effective than the sole use of Bu_4NBF_4 (entries 1, 2), thereby providing evidence for the crucial role of acidic conditions. A similar effect can also be observed for Bu_4NOAc and Bu_4NI (entries 3-6). The presence of the acid HOAc (entry 7), which is more soluble in MeCN compared to KHSO_4 , causes the decomposition of the sulfate product **1**. Additionally, side products such as methyl acetate and acetylated-**1** derivatives are formed, resulting in a lower overall yield.

Entry	Activator	Yield of 1 (%) ^a
1	Bu_4NBF_4	21%
2	Bu_4NBF_4 , KHSO_4	62%
3	Bu_4NI	49%
4	Bu_4NI , KHSO_4	66%
5	Bu_4NOAc	58%
6	Bu_4NOAc , KHSO_4	79%
7	Bu_4NOAc , HOAc	30%

^a Yields were determined by ^1H NMR using dibromomethane as an internal standard.

We have updated these data in revised Supporting Information (page 5).

2. On the other hand, the yield with a counter anion other than HSO_4^- is quite good for entries 4-6, especially for entry 5, even in the absence of acid. Does the author have an explanation for how the product was formed in this case?

We appreciate the reviewer's meticulous analysis of this aspect. The potential release of conjugated acid (HX) from the equilibrium between the substrate ROH and activator Bu_4X ($\text{X} \neq \text{HSO}_4^-$) may indeed promote sulfation. This hypothesis can be verified by the decreased pH over time for the solution of **1a** and Bu_4OAc .

These results can be found in revised Supporting Information (page 6):

pH over time for the mixture of **1a** and Bu_4NOAc (1.0 equiv) in CH_3CN (1M) at 80 °C

Time	pH indicator paper	pH meter
1 h	6.7	11.38
3 h	6.4	10.22
7 h	6.2	10.02
12 h	6.0	9.01

In the revised manuscript, we have added the discussion:

“The use of Bu_4NBF_4 , Bu_4NOAc and Bu_4NI as additive that could potentially release their conjugated acid from the equilibrium with **1a**, resulted in lower yield.”

Substrates scope:

3. All the products were well-characterized; however, only NMR yield was provided for substrates (11, 19-24, 58, 61, 65, 66).

Addressed. Please see our reply to comment 1 (on SI) of reviewer 1.

4. For conditions d in Table 2, several other counter cations like Na⁺, K⁺, were used. How did the authors determine that these were not included in the final products?

We thank the reviewer for this concern. In our case, organic sulfates with different cations have different polarity. We choose compound **1** as an example, and it is evident that they are well separated on the TLC plate. Compared to Na⁺ and K⁺, the soft Bu₄N⁺ cation are considered to be more matched to organic sulfate anion, and their corresponding sulfates exhibit better solubility.

Mechanistic studies:

5. In Figure 2 A, the ¹H NMR spectrum of methyl monosulfate (1-B) does not appear consistent with it being the main intermediate, especially given its low integration. Without integrations of the key signals, it is hard to discern the correlation between the new peaks in the ¹H NMR and that they correlate to the main intermediate. Integration of these peaks in relation to the standard in ¹H spectra would be helpful.

We agree that it would be appropriate to give the integration of the intermediates. The new data is added into Figure 2A in the revised manuscript.

In the revised manuscript, we add the discussion about the mechanism:

“As the reaction progressed, these intermediate species were converted to some extent into the final sulfate product.”

6. According to the proposed mechanism in Figure 1C-ii, Bu₄NHSO₄ would react with intermediate B. Why wouldn't it react with DMS at the beginning, given that DMS is less sterically hindered than intermediate B? We agree with the reviewer, and have corrected/removed the inappropriate expression in Figure 1C-ii. A_{SU}2 mechanism starts with the activation of DMS in the presence of Bu₄NHSO₄, we have corrected it in the revised manuscript in Figure 1C-ii:

(C) Nucleophilic substitution on DMS by B_{AL}2 and A_{SU}2 mechanisms

7. Have the authors investigated the formation of SO₃ during the reaction? They could attempt to detect and capture SO₃ by mixing the DMS and Bu₄NHSO₄. This would enhance the understanding of the reaction mechanism.

Addressed. Please see our reply to comment 15 of review 1.

8. Determining the source of the 'sulfur' in the product is crucial, and conducting experiments with ³⁴S labeled Bu₄NHSO₄ and/or DMS would provide valuable insights.

We agree with the reviewer that establishing the source of sulfate in the product is essential for gaining mechanistic insights. The ³⁴S-labeling experiment provides direct evidence, but synthesizing ³⁴S-labeled Bu₄NHSO₄ and DMS from H₂³⁴S or ³⁴S is challenging due to their high cost and limited availability. Here, we present two experiments to support the sulfate originating from both of DMS/DPS, and Bu₄NHSO₄. First, the yield of sulfate product **1** exceeds the stoichiometric amount of any possible sulfur source. Second, DMS reacts with Bu₄NHSO₄ to yield methyl sulfates mixture (**D** and **D'**), which could deliver the final organic sulfates.

(C) Stoichiometric O-sulfation

(B) Formation of **D** and **D'**

(E) Reactivity of **D'**

In the revised manuscript, we add the following discussion:

“The higher yield of **1**, in comparison to the stoichiometric amount of any individual sulfur source, suggests that the sulfate group in product originates from both DMS/DPS, and $\text{Bu}_4\text{NH}_4\text{SO}_4$ (Figure 2C).”

9. In drawing the mechanism (Figure 1C-ii, from A to B), based on the arrows the authors have depicted, it suggests the formation of a methoxide anion. However, considering the acidic conditions, the existence of a methoxide anion is improbable. The arrow might be better represented as originating from the oxygen of the alcohol rather than the hydrogen.

We thank the reviewer for pointing out this error. We thank the reviewer for pointing out the confusion of the arrows caused. As reviewer suggested, we have corrected it in the revised manuscript:

SI and others:

10. It appears that the R_f values for most products are identical. Could the authors please confirm this observation? Additionally, it's noted that most of the products share the same physical state, described as a light yellow oil.

We thank the reviewer for pointing out this irregularity of R_f values. The polarity of organic ionic sulfates predominantly depends on the sulfate (SO_4^-) group. Due to their relatively lower solubility in the organic mobile phase (DCM/MeOH), the individual organic sulfates often appear as trailing spots. Thus, most of these organic sulfates exhibit nearly identical R_f values.

The discrepant R_f values we observed have been corrected in the revised Supporting Information.

Compound **4**: revised R_f is 0.30 (previously 0.40).

Compound **40**: revised R_f is 0.30 (previously 0.40).

Compound **61**: revised R_f is 0.20 (previously 0.40).

Compound **65**: revised R_f is 0.20 (previously 0.40).

We have updated the information on physical state for compound **8**, **25**, **27**, **33** and **65** in the revised Supporting Information. For example, Page S11: “2,6-dichlorophenethanol sulfate **8** was purified by flash chromatography on silica gel eluting with DCM/MeOH (100:1 to 20:1, v/v) to afford the product as a **white solid** (81.8 mg, 0.160 mmol, 80%).”

11. The abbreviation for diisopropyl sulfate is not consistent between the main text and the SI. Please ensure uniformity in the usage of abbreviations throughout the manuscript.

We thank the reviewer for pointing out these errors. We have scrutinized the manuscript and supplementary information to use “DPS” as the abbreviation of diisopropyl sulfate.

12. Some typos: Figure 1C-i -> Figure 1C-ii; line 93: 12 h -> 12 hours; lines 95-98: entry -> entries; line 97: slower yield -> lower yield.

We thank the reviewer for pointing out these errors. The corrections are reflected in the revised manuscript and supplementary information.

We are grateful to the reviewers for the thoughtful comments and feel that we have been able to address them, which resulted in a substantially improved manuscript.

REVIEWERS' COMMENTS

Reviewer #1 (Remarks to the Author):

I thank the authors for addressing all concerns raised and am now satisfied with the responses and look forward to seeing the work online in the future.

Reviewer #2 (Remarks to the Author):

The original manuscript was carefully revised, and the comments of this and all other reviewers were answered in detail and incorporated into the revised manuscript. The article can therefore be published in this form.

Reviewer #3 (Remarks to the Author):

Having seen an earlier version of this manuscript, I am very happy with the response of the authors to my comments and to the comments of the other reviewers. The authors fully addressed the review comments, especially, the mechanistic studies were improved on the basis of several control studies in the revised manuscript and supplementary information

Although the ^{34}S labeled experiments were not conducted due to synthetic challenges, the alternative stoichiometric experiment provided clear evidence that the 'sulfur' could originate from both DMS/DPS and Bu_4NHSO_4 . I commend the authors for attempting to detect the formation of SO_3 ; however, it appears that an SO_3 complex was not formed during the reaction. I am certainly now happy to recommend publication, however, there are still a few minor issues that I feel need to be addressed:

1. In Figure 1C-ii, based on Figure 2E, it appears that D' leads to the product. Base on this, D' itself or its variants could be also serve as the O-sulfation reagents with less toxicity. Additionally, could the authors investigate whether D could result in the formation of the product?
2. In Figure 2B, it would be beneficial to specify which compound is formed in the 93%.
3. For Figure 2B-2F, it would be better to include the reaction time in the schemes.

An impressive study of a highly useful process.

Referees' comments:

Reviewer 1 (Remarks to the Author):

I thank the authors for addressing all concerns raised and am now satisfied with the responses and look forward to seeing the work online in the future.

We are grateful to hear from the reviewer that we are on the right track.

Reviewer 2 (Remarks to the Author):

The original manuscript was carefully revised, and the comments of this and all other reviewers were answered in detail and incorporated into the revised manuscript. The article can therefore be published in this form.

We are grateful to hear from the reviewer that we are on the right track.

Reviewer 3 (Remarks to the Author):

Having seen an earlier version of this manuscript, I am very happy with the response of the authors to my comments and to the comments of the other reviewers. The authors fully addressed the review comments, especially, the mechanistic studies were improved on the basis of several control studies in the revised manuscript and supplementary information

Although the ³⁴S labeled experiments were not conducted due to synthetic challenges, the alternative stoichiometric experiment provided clear evidence that the 'sulfur' could originate from both DMS/DPS and Bu₄NHSO₄. I commend the authors for attempting to detect the formation of SO₃; however, it appears that an SO₃ complex was not formed during the reaction. I am certainly now happy to recommend publication, however, there are still a few minor issues that I feel need to be addressed:

1. In Figure 1C-ii, based on Figure 2E, it appears that **D'** leads to the product. Base on this, **D'** itself or its variants could be also serve as the O-sulfation reagents with less toxicity. Additionally, could the authors investigate whether **D** could result in the formation of the product?

We greatly appreciate the reviewer's valuable input regarding the sulfation reactivity of **D** and **D'**. Indeed, we have investigated **D'** and its variants, such as **D''**, for sulfation on several substrates. Unfortunately, we observed that the yields obtained using these derivatives were significantly lower than those obtained with DMS at the current stage.

When considering **D** with H⁺ as a cation, it is important to note that **D** only achieves sulfate product **1'** in 7% yield. In contrast, the presence of Bu₄NHSO₄ significantly enhances the yield to 70%. This stark difference further emphasizes the significance of the activation system (Bu₄NHSO₄) in our sulfation process.

Reactivity of **D**

Therefore, our next objective is to explore new activation systems that can achieve sulfation with higher efficiency and broader substrate scope for **D'** and **D**, as well as their variants.

We have updated these data in revised Supporting Information (page 60).

2. In Figure 2B, it would be beneficial to specify which compound is formed in the 93%.

We apologize for our initial inability to provide a detailed explanation for Figure 2B. Upon reaction of DMS with Bu₄NHSO₄ (as depicted below), a mixture of **D** and **D'** products is formed, with neither **D** nor **D'** being the sole product. Theoretically, 1 equivalent of DMS can afford 2 equivalent of **D** and **D'** mixture, which can't be differentiated by ¹H NMR. Consequently, it is not possible to determine definitively which of the two compounds is formed in this reaction.

In the revised manuscript, we have made the necessary corrections to Figure 2B to ensure clarity and understanding for the audience:

(B) Formation of **D** and **D'** mixture

3. For Figure 2B-2F, it would be better to include the reaction time in the schemes.

An impressive study of a highly useful process.

We thank the reviewer to pointing this out, and include the reaction time in the revised schemes:

(B) Formation of **D** and **D'** mixture

(C) Stoichiometric O-sulfation

(D) Reactivity of **1-B**

(E) Reactivity of **D'**

(F) ¹⁸O-labeling experiment

We are grateful to the reviewers for the thoughtful comments and feel that we have been able to address them, which resulted in a substantially improved manuscript.